# OKR-Agent: An Object and Key Results Driven Agent System with Hierarchical Self-Collaboration and Self-Evaluation

## Abstract

In this study, we introduce the concept of OKR-Agent designed to enhance the capabilities of Large Language Models (LLMs) in task-solving. Our approach utilizes both self-collaboration and self-correction mechanism, facilitated by hierarchical agents, to address the inherent complexities in task-solving. Our key observations are two-fold: first, effective task-solving demands in-depth domain knowledge and intricate reasoning, for which deploying specialized agents for individual sub-tasks can markedly enhance LLM performance. Second, task-solving intrinsically adheres to a hierarchical execution structure, comprising both high-level strategic planning and detailed task execution. Towards this end, our OKR-Agent paradigm aligns closely with this hierarchical structure, promising enhanced efficacy and adaptability across a range of scenarios. Specifically, our framework includes two novel modules: hierarchical **O**bjects and **K**ey **R**esults generation and multi-level evaluation, each contributing to more efficient and robust task-solving. In practical, hierarchical OKR generation decomposes Objects into multiple sub-Objects and assigns new agents based on key results and agent responsibilities. These agents subsequently elaborate on their designated tasks and may further decompose them as necessary. Such generation operates recursively and hierarchically, culminating in a comprehensive set of detailed solutions. The multi-level evaluation module of OKR-Agent refines solution by leveraging feedback from all associated agents, optimizing each step of the process. This ensures solution is accurate, practical, and effectively address intricate task requirements, enhancing the overall reliability and quality of the outcome. Experimental results also show our method outperforms the previous methods on several tasks.

## 1 Introduction

The widespread application of Large Language Models (LLMs) has elicited transformative advancements in various sectors. However, the intricate potential of LLMs remains underexplored, especially in tasks of higher complexity Qin et al. (2023); OpenAI (2023), such as curating movie scenes or designing sophisticated travel plans, where LLMs face challenges related to knowledge and reasoning intensity, due to issues like hallucination ( Bang et al. (2023); Bubeck et al. (2023))and lack of slow thinking Sloman (1996); Lin et al. (2023a).

In this study, we explore two crucial dimensions of utilizing LLMs for intricate tasks: enhancing self-collaboration and enabling self-evaluation. Our investigation is motivated by two key observations: firstly, existing studies like CoT Wei et al. (2023) and ToT Yao et al. (2023a) reveal that introducing intermediate steps and adopting a multi-persona approach can markedly enhance LLM performance. Nevertheless, these methodologies necessitate manual workflow configuration and agent assignment. The study SPP Wang et al. (2023) demonstrates the potential of LLMs to dynamically allocate agents, based on task inputs and user-specified requirements, and to produce rational outputs. This led us to explore deeper into the capability of LLMs to autonomously decompose input tasks into meaningful goals and formulate guidelines for agent collaboration from varied perspectives. Secondly, the inherent multiplicity of potential solutions in any collaborative undertaking necessitates the critical ability to discern, assess, and choose the most apt solutions Li et al. (2023). Hence, we aim to unearth

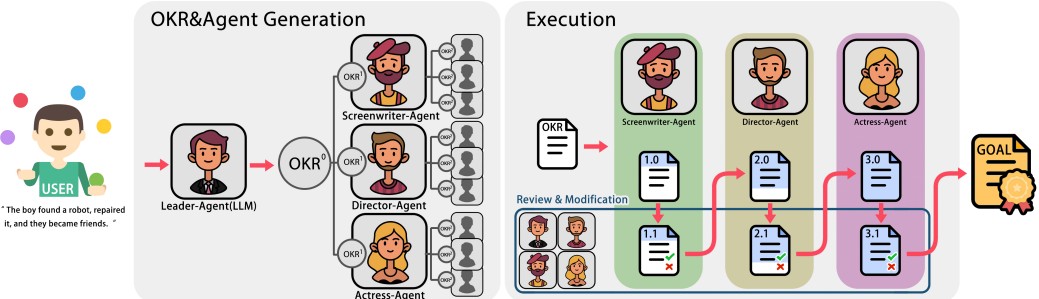

Figure 1: Pipeline of OKR-Agent: we utilize a hierarchical approach, where agent(LLM) decomposes tasks into Objectives and Key Results, spawns role-specific agents, and fosters collaboration and evaluation.

a mechanism that enables LLMs to effectively evaluate and pinpoint the most promising solutions from a myriad of generated possibilities.

Inspired by the renowned success of the Objectives and Key Results (OKR) system in guiding large corporations and organizations towards achievement, we have developed a unique goal-setting framework within our Large Language Model (LLM) task-solving pipeline,namely OKR-Agent. Specifically, OKR-Agent works in a hierarchical and self-collaboration manner: Given a concise description of a task, a primary agent undertakes initial analysis and generates a spectrum of potential Objectives, proceeding to select the most apt ones. Subsequently, this leading agent incorporates additional agents based on the finalized objectives, assigning to each the corresponding Key Results necessary for fulfillment. It is imperative to note that this process of agent assignment is comprehensive, encompassing both the delineation of roles for each agent and the establishment of inter-agent dependencies throughout the workflow. These newly incorporated agents possess the capability to further break down the Key Results into subordinate objectives and key results, enabling a substantial enlargement of the agent roster. This iterative, multi-level approach ensures that each layer of the task has a dedicated focus, fostering a nuanced and thorough exploration of potential solutions and strategies.

Another core component of OKR-Agent is the multi-level self-evaluation. As illustrated in recent works Park et al. (2023); Hong et al. (2023), assimilating evaluations from varied personas and perspectives substantially enhances the accuracy and quality of outputs generated by LLM agents. However, prevailing methodologies solicit feedback from agents based solely on their designated roles, often neglecting a holistic overview of the content. We postulate that an effective agent evaluation should not only be reflexive but also encompass assessments from agents in close relational proximity, offering a more rounded perspective. With the architectural design of OKR-Agent, each agent is cognizant of its relative position within the workflow, enabling it to furnish evaluations that encompass all correlated perspectives and make modifications accordingly. Moreover, proficient evaluations at both strategic and executional levels are crucial to guarantee the efficacy of the solution. The self-evaluation works in a 'coarse-to-fine' manner, where top-level agents scrutinize overarching strategies, subsequently transitioning to lower-level agents who meticulously attend to executional details.

We further validate the efficacy of OKR-Agent through experiments encompassing three diverse tasks: short video storyboard generation , multi-day trip planning, and trivia creative writing. The empirical results denote that OKR-Agent surpasses preceding LLM-based task-solving methodologies, exhibiting superior performance in both overarching planning and the intricacy of details, presenting a consistent enhancement across varied domains.

To summarize, our main contributions are as below:

- We introduce a new hierarchical and self-collaborative approach to task-solving. It analyzes and decomposes tasks into distinct Objectives and assigns Key Results to various agents, based on their roles and the workflow's relative positions, enabling a more structured and coherent approach to task execution.

- We propose a novel multi-level self-evaluation mechanism, allowing each agent to offer evaluations from all related perspectives. This feature not only refines the accuracy and quality of the outputs by incorporating varied assessments and feedback but also ensures that the evaluations are comprehensive, covering both strategic and executional levels.

- OKR-Agent has demonstrated its supremacy over existing LLM-based task-solving models on on diverse tasks. It has shown consistent enhancements in both overall planning and detail execution, making it a robust solution for complex task-solving scenarios.

## 2 RELATED WORK

### 2.1 PROMPTING FRAMEWORK AND PIPELINE

Wei et al. (2023) introduces the concept of Chain-of-Thought, which effectively enhances the reasoning ability of LLMs by generating a series of intermediate reasoning steps in response to a question. Yao et al. (2023a) proposes to explore multiple feasible paths and combine searching and backtracking, which significantly improves the effectiveness in solving complex logical reasoning problems. Wang et al. (2023) proposes a reasoning method with automatic evaluation. This approach involves automating the setup of multiple agents with different capabilities to evaluate and refine reasoning results multiple times. It endows LLM with stronger reasoning abilities while effectively reducing hallucinations. Besta et al. (2023) introduces a GoT structure that can comprehensively utilize the optimal results generated during the reasoning process. While ongoing advancements aim to augment the task-solving capabilities of LLMs through the implementation of diverse reasoning pipelines, a notable performance gap persists, particularly in generating intricate content requisite for domains like creative writing and storyboard generation. In this study, we explore an innovative pipeline imbued with a hierarchical structure specifically engineered to address the complexities inherent in such creative generation tasks.

### 2.2 SPECIFIC TASKS-SOLVING WITH AGENT AND LLM

Numerous studies have delved into the enhancement of LLMs' task-solving capabilities, exploring innovative paradigms to boost their creative and problem-solving prowess in specific scenarios. Studies like Li et al. (2023) introduced systems like CAMEL, optimizing inter-agent communication, whereas others like Park et al. (2023) harmoniously amalgamated the behavioral propensities of agent groups with the advanced reasoning capabilities of LLMs. These explorations, inclusive of works cited as Hong et al. (2023); Cai et al. (2023); Lin et al. (2023b), have manifested remarkable advancements in tailoring Agent-Pipeline solutions, demonstrating unparalleled creativity and proficiency in problem resolution. Conversely, research ventures like Mirowski et al. (2023) have employed hierarchical, structured approaches combined with specific role assignments to leverage the capabilities of LLMs in generating long-form creative content, such as continuous scripts enriched with detailed contextual elements. These studies, along with Zhang et al. (2019); Mishra et al. (2023); Liu et al. (2023a), have exemplified commendable strides in integrating domain-specific knowledge to guide LLMs in executing tasks with enhanced precision in respective domains.

Despite the plethora of advancements and innovations in LLMs, a common limitation is evident—the reliance on manual specification for both problem-solving processes and the determination of agent attributes, impeding their versatility in generalized applications. To address this, our study proposes leveraging LLMs as agents capable of self-collaboration and self-evaluation, a methodology we posit is adaptable across a myriad of tasks.

### 2.3 COGNITIVE SCIENCE AND LLM

Researchers such as Piaget (1954); Pellegrini (2009); Wason & Johnson-Laird (1972); Sloman (1996) have delved into the realms of human psychology and cognition, influencing subsequent developments in artificial intelligence theory Chandrasekaran et al. (2017). Devlin et al. (2019); Brown et al. (2020); OpenAI (2023); Chowdhery et al. (2022); Srivastava et al. (2023) demonstrated the capabilities of large language models (LLMs) to the public. Shuster et al. (2022); Bang et al. (2023); Liu et al. (2023b); Yao et al. (2023b); Lin et al. (2023a); Madaan et al. (2023); Shinn et al.

(2023), by integrating cognitive science with Large Language Models (LLMs), continuously explore methods to enhance the capabilities of LLMs.

All of these approaches represent new explorations in both theoretical and methodological aspects, providing fresh perspectives for the enhancement and development of LLMs in the future.

## 3 METHOD

In this section, we formally introduce OKR-Agent, as demonstrated in Figure 1 . We first revisit the definition of task-solving of LLM Agents, and then provide details about OKR-Agent including Self-Collaboration, Self-Evaluation, and the complete workflow.

Given an input instruction $x$ and a model $\mathcal{M}$, if we denote the final output to be $y$, then the Standard Prompting can be formulated as:

$$y = \mathcal{M}(x)$$

with the additional prompt $p$ and intermediate generations $z$, Chain-Of-Thought(CoT)and Solo-Performance-Prompting(SPP) can be described as below respectively:

$$y = \mathcal{M}(p_{cot}|x|\{z_1, z_2, ..., z_n\})$$

$$y = \mathcal{M}(p_{spp}|x|z_p|\{z_b^1, z_b^2, ..., z_b^n\}|\{z_s^0, z_f^1, ..., z_f^m\}_{j=1,2,..n})$$

where $z_i$ is the intermediate step in CoT, $z_p, z_b, z_f$ are the multi-personas and multi-run feedback in SPP. For OKR-Agent, the goal is to hierarchically generate Objects($z_o$) and Key Results($z_k$) from user input $x$, and assign persona $z_p$ accordingly. We also want the system can generate evaluations($z_e$) as guidance during execution. Our system then can formulated as:

$$y = \mathcal{M}(p_{okr}|x|z_p|\{z_o^1, z_o^2, ..., z_o^n\}\{z_k^1, z_k^2, ..., z_k^n\}\{z_e^1, z_e^2, ..., z_e^n\}|_{j=1,2,..n}) \qquad (1)$$

We now provide the details of the corresponding intermediate generations ($z$) in OKR-Agent.

**Objects($z_o$) and Key Results($z_k$).** At the core of our OKR-Agent pipeline is the hierarchical derivation of Objects and Key Results. Given a user input $x$, the LLM acts as a Leading Agent (LA), assigned to produce a set of potential targets that align with the user's intentions. This generative process can occur multiple times, allowing the LA to consolidate a selection of the most relevant targets, referred to as the first level of Objects ($z_o$). Subsequently, each object can be elaborated into sub-objects, providing more nuanced details and forming the Key Results $z_k$.

**Agent Assignment($z_p$).** Given the generated Objects($z_o$) and Key Results($z_k$), each can be paired with an agent, constituting a structured set of agents. It is noteworthy that, unlike the SPP in Wang et al. (2023), our agents are not chosen from a predefined list but are dynamically incorporated based on OKR decomposition. This approach ensures that agent selection is not only task-driven but also inherently adaptive to the tasks at hand. Specifically, each agent is accountable for its assigned Object or Key Result: agents assigned to Objects oversee the broader aspects of the task, ensuring overarching coherence, while those paired with Key Results focus on the fine details, maintaining precision in execution.

**Evaluations($z_e$).** For each agent, it is crucial to develop evaluation criteria, represented as $z_e$, to direct its work. Once again, we employ LLM for criteria generation. Given the role definition and corresponding OKR for an agent, we prompt the LLM to formulate a concise, one-sentence criterion serving as the evaluation metric for the relevant segment in the final output.

### 3.1 HIERARCHICAL GENERATION

As depicted in Algorithm 1, the decomposition process of OKR, along with the generation of agents and evaluations, can be conducted iteratively, allowing for a meticulous and organized delineation of tasks, which ensures accuracy and uniformity throughout the task-solving trajectory. Formally, the output of this generation stage is the structured OKR $z_o, z_k$ with its associated agents $z_p$ and the evaluations $z_e$. This hierarchical configuration intrinsically forms inter-dependencies among agents, enabling streamlined and efficient pathways for execution and evaluation.

---

**Algorithm 1 GenOKR(UserInput, Hierarchy)**

---

OKR = [], Agents = [], Evaluations = []
OKRTmp = [UserInput]
**for** level in Hierarchy **do**
    levelOKRs = Generate Objects and Key Results From OKRTmp using LLM
    levelAgents = Generate Agents From OKRs using LLM
    levelEvaluations = Generate Evaluations From OKRs using LLM
    OKR += levelOKRs, Agents += levelAgents ,Evaluations += levelEvaluations
    OKRTmp = levelOKRs
**end for**
return OKR, Agents, Evaluations

---

**Generation Prompt.**   To prompt a LLM to follow the generation procedure as mentioned above, we also designed the prompts of OKR decomposition and evaluation criteria, which help LLM to perform as expected. Specifically, we use the following for OKR decomposition:

**"You are an expert in 'Objective templates'. It is known that a key element of the 'Objective template' includes 'KR0,KR1,...,KRn'. Expand the keywords 'KRx' within this Objective. List keywords and separate each word with a comma, no extra words."**

For agent generation, we use:

**You are an expert in 'Objective', tasked with creating content about 'KR0,KR1,...,KRn'. Summarize the required job positions as 'JobTitle'. Connect the 'JobTitle' with '->', without unnecessary words.**

Moreover, we use this prompt to generate evaluation criteria:

**You are an outstanding 'JobTitle Expert'. Provide a sentence that thoroughly describes the role of this position in 'Objective' work, taking into account the professional characteristics of 'JobTitle'. The sentence should follow the structure "An excellent 'Objective' should have the characteristic of... in the aspect of XXX;" and should not exceed 200 words. Avoid unnecessary words.**

### 3.2   OKR-AGENT WORKFLOW

Following the **Hierarchical Generation** stage, the LLM advances to the execution phase. In this phase, each agent is tasked with extending the solution per its assigned OKR, scrutinizing the extended solution against its evaluation criteria and the accumulated ones from preceding agents, and refining the solution based on the evaluations before forwarding the refined version to the subsequent agent. This progression operates hierarchically, cascading from the Leading Agent down to the leaf-node agents, culminating in the formulation of the final solution. We now provide details for each component, the overall workflow is shown in Algorithm 2.

**Solution expanding.**   For each agent, the essential task is to elaborate on the existing solution per its designated OKR, which may involve enumerating sub-targets or enriching the details of key results. Given that we employ OKR decomposition for the input tasks, it offers an intuitive **key-value** method to represent the solution, wherein the keys can serve as objects and the values can retain the details. In our experiments, we discovered that such a representation is more conducive for LLMs in maintaining complicate information. We will provide examples in experiment section.

**Solution Review.**   Once the solution is expanded, the subsequent step for the agent is its review. Our findings indicate that errors introduced at higher levels tend to propagate and accumulate in subsequent levels, making it challenging for later agents to rectify them. Therefore, post-generation review by the agent becomes imperative. To safeguard previously generated parts from unintended alterations, we catalog the evaluation criteria of all preceding agents. These criteria are then amalgamated with the current agent's criteria, enabling a more precise and comprehensive evaluation.

---

**Algorithm 2 OKR-Agent Workflow**

---

    Answer = Null
    OKR,Agents,Evaluations = **GenOKR(UserInput)**
    WorkingEvaluation = []
    **for** Agent in Agents **do**
        WorkingEvaluation.append(Evaluations[Agent])
        AnswerTemp = Agent update Answer with its OKR.
        ReviewResult = Using WorkingEvaluation to get feedback for AnswerTemp
        AnswerTemp = Modify AnswerTemp with ReviewResult
        Answer = AnswerTemp
    **end for**
    return Answer

---

**Solution Modification.** Given the current solution and evaluation, the agent will be asked to modify the solution in case there are errors, missing parts and so on. This refinement process can be executed multiple times, allowing the agent to choose the iteration with the highest evaluation score.

## 4 EXPERIMENTS

In this section, we deploy the OKR-Agent to three distinct tasks: storyboard generation, creative writing, and trip planning, to evaluate its efficacy and versatility in varying applications. Across these diverse tasks, OKR-Agent exhibited exceptional capabilities in global task planning, maintaining high levels of correctness, and generating rich details.

### 4.1 SHORT VIDEO STORYBOARD GENERATION

**Storyboard generation.** In the creation of a short video, a storyboard plays a pivotal role by taking a writer's narrative and structuring it for video production, delineating essential components like shot composition, scene setup, actor performance, and dialogue through textual representations. It acts as the linchpin, coordinating tasks across various roles in production. Nonetheless, crafting a proficient storyboard script necessitates extensive professional knowledge. To democratize the creation of visually compelling videos and make it accessible to everyone, we explored the proficiency of OKR-Agent in executing this intricate task.

**OKR Generation.** In our experiment, we used the following user input: "A storyboard of 'The boy picked up a robot, the boy repaired the robot, the boy and the robot became friends'." Referencing 2, we contrast the Objects and Key Results produced by OKR-Agent and ChatGPT. Owing to its hierarchical structure, OKR-Agent elucidates more detailed and significant elements and requirements for video production than ChatGPT. Additionally, our method autonomously generates the requisite agents, deriving from the OKRs; for instance, the agents for OKR-L-2 include 'script writer', 'director', 'camera operator', 'actor', and 'musician'.

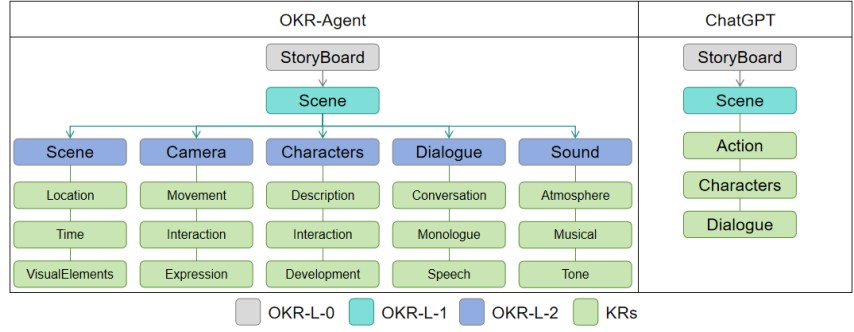

Figure 2: OKR-Agent vs ChatGPT

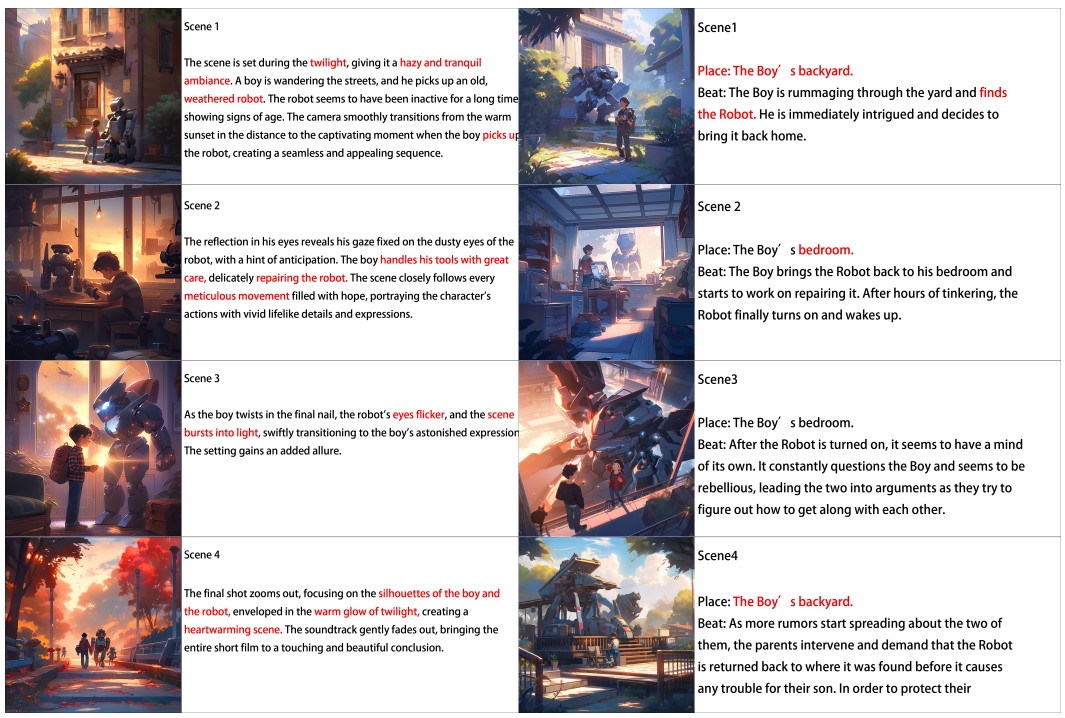

Figure 3: Storyboard visualization. Left: OKR-Agent; Right: Dramatron.

**Storyboard visuallization.** Given the substantial visual and artistic elements in the storyboard produced by OKR-Agent, we proceeded with an evaluation, employing user-study based subjective assessments. We initiated the evaluation by generating results from OKR-Agent using the uniform story input ("The boy found a robot, fixed it, and they became friends.") and visually represented the script content utilizing AI "text to image" tools like MidJourney Midjourney.com (2023), where the generated outputs served as prompts for deriving results. For comparative analysis, results were similarly procured using DramaTron Dramatron (2023), with the comparative outputs illustrated in Figure 3

We subsequently conduct a user-study involving 30 participants, comprised of 20 professionals from the field of 'Digital Art Creation' and 10 non-professionals. Participants were queried on various aspects including 'Plausibility of the Story,' 'Text/Image Consistency,' and 'Visual Continuity,' with the responses statistically illustrated in Table 1. The compiled data revealed that OKR-Agent, leveraging its enhanced capability for text detail generation, demonstrated superior control over visual elements in the text-to-image transformation process, marking a 24% improvement, and yielded more consistent results, with a 28.9% increase compared to Dramatron. However, Dramatron exhibited a richer storyline content, surpassing OKR-Agent by 5.2%, attributed to its distinctive ability to define 'storyline styles.' The comparison between 'Professional' and 'Amateur Evaluation' discernibly shows that professionals applied stricter assessment criteria, especially evident from Dramatron's scores. However, this stringent assessment did not translate to substantial differences in the evaluation of OKR-Agent, underscoring the resilience and efficacy of OKR-Agent's detailed output. More visualization results of another input is provied in 4.

## 4.2 MULTI-DAY TRIP PLANNING

This task aims to assess the coordination of individual sub-item arrangements within OKR-Agent in multiple parallel planning scenarios, as well as the continuity and rationality of global planning. This feature is particularly prominent in real-world travel planning tasks. Such planning is conducted on a 'day'-by-'day' basis, where multiple factors need to be taken into account within a single day, each with various inter-dependencies. Simultaneously, the overall arrangement also needs to be considered for its rationality.

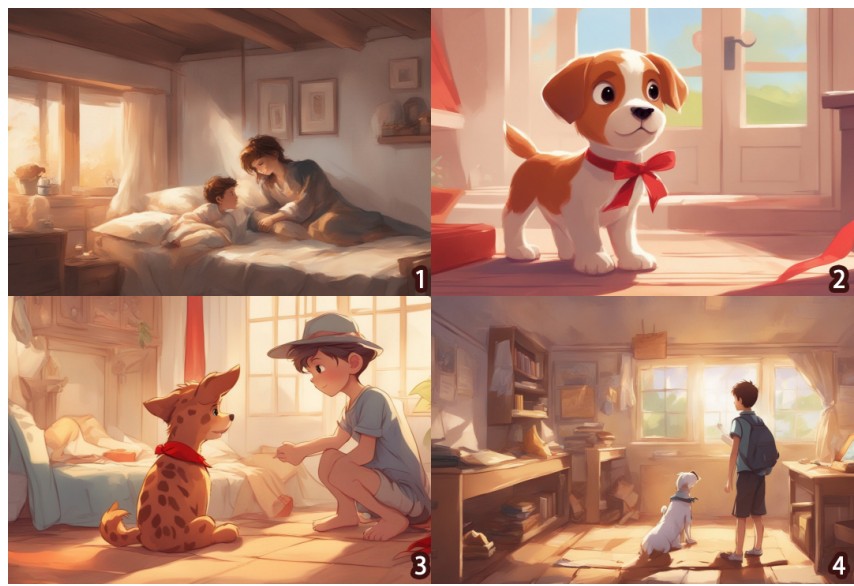

Figure 4: Another Story:Mom is making breakfast and calls for the boy to wake up. The boy lingers in bed. The puppy rushes in and wakes him up. The boy grabs his backpack and heads to school.

| | | Plausibility of the Story | Text/Image Consistency | Visual Continuity | Average | Advantages | Weaknesses |
|---|---|---|---|---|---|---|---|
| OKRAgent | Prof. | 3.24 | 3.61 | 3.34 | 3.34 | Rich in details. | The story may appear a bit flat. |
| | Amtr. | 3.25 | 3.75 | 3.5 | 3.45 | | |
| | W.M. | 3.24 | 3.65 | 3.39 | 3.37 | | |
| Dramatron | Prof. | 3.3 | 2.75 | 2.48 | 2.875 | The story is vivid. | The coherence between text and visuals is weak |
| | Amtr. | 3.7 | 3.3 | 2.95 | 3.25 | | |
| | W.M. | 3.43 | 2.93 | 2.63 | 3 | | |

Table 1: Subjective Evaluation of Comparative Experiment on Text-to-Image Generation

In this task, OKR-Agent breaks down the mission into four objectives: determining the travel destination and itinerary, booking transportation and accommodation, planning daily activities, and preparing contingency plans. This results in the corresponding agents: 'Travel Planner', 'Accommodation and Transportation Booking Officer', 'Emergency Management Commissioner', responsible for 'itinerary arrangement', 'transportation and accommodation planning', and 'safety and financial management tasks' respectively.

Table 2 illustrates the divergent output results from OKR-Agent and ChatGPT, both generated from identical user input—'Family Three-Day Hawaii Travel Plan.' OKR-Agent exhibits a more thorough consideration of elements such as 'transportation' and 'financial management' compared to ChatGPT, which is evidenced by the detailed inclusion of aspects like 'round-trip time arrangements for the first and last days' and 'airfare and car rental costs' in the planning results. This reflects OKR-Agent's superior capability in producing comprehensive and realistic travel plans, catering to multiple facets of trip planning.

### 4.3 TRIVIA CREATIVE WRITING

Creative Writing is the art of crafting written content that explores thoughts, feelings, and ideas, utilizing narrative craft, character development, and literary tropes, often diverging from formal writing styles to convey emotions, create imagery, and experiment with language across various genres and forms like novels, poetry, and screenplays. In this task, we test the same input as in the SPP paper: "Generate a paragraph of fantasy creative story."

| | Day 1 | Day 2 | Day 3 |
|---|---|---|---|
| OKR | The destination is the famous Waikiki Beach, where visitors can enjoy traditional Hawaiian performances in the evening. The mode of transportation is by plane. Local cuisine includes traditional Hawaiian BBQ and seafood. As for the budget, transportation is estimated at 1000 USD, dining at 200 USD, and attraction tickets at 100 USD. Accommodation has been arranged at a luxury beachfront hotel with amenities like a swimming pool and gym. The itinerary also covers taxi transportation to the beach, sightseeing activities, and dining arrangements by the beach. | This one-day itinerary revolves around a visit to the Volcanoes National Park in Hawaii. The mode of transportation is by renting a car, allowing flexibility in exploration. A must-try local delicacy is the Hawaiian volcano stone-grilled chicken. The day's activities include touring the Volcanoes National Park and experiencing its unique volcanic terrain. Dining arrangements offer the choice of dining with the tour group or sampling snacks at the park. The estimated budget covers transportation at 100 USD, dining expenses of 40 USD for lunch and 50 USD for dinner, and attraction tickets at 30 USD. Accommodation is arranged at a luxurious beachfront hotel with amenities like a swimming pool and gym, situated conveniently by the beach. | This half-day itinerary will take you to Pearl Harbor in Hawaii. The mode of transportation is by taxi, which is convenient and efficient. Local delicacies such as American hamburgers, offering delicious options. The itinerary includes taking a taxi to Pearl Harbor, participating in sightseeing activities at Pearl Harbor, and shopping. Dining arrangements will be made at restaurants within the tourist area for a convenient experience. In terms of budget, transportation is estimated at 70 USD, dining expenses include lunch at 30 USD and dinner at 50 USD, and attraction tickets are estimated at 20 USD. Accommodation is arranged at a luxurious beachfront hotel, conveniently located by the beach, and offers amenities such as a swimming pool and gym. In the evening, prepare to back home by plane. |
| ChatGPT | On the first day, you will step onto the golden sands of Waikiki Beach, feeling the deep blue sea and the cheerful waves. You can choose to surf, take a leisurely stroll, or simply lie on the beach and bask in the sunlight. Come lunchtime, try the local delicacies and savor fresh seafood and tropical fruits. As evening approaches, opt for a meal at a beachside restaurant and enjoy the sunset casting its glow on the water's surface. | On the second day, you will head to the Volcanoes National Park to explore the magnificent landscapes of an active volcano. You can embark on a hiking adventure to witness the awe-inspiring sight of flowing lava, or visit museums to understand the formation process and history of volcanoes. For lunch, you can choose to dine at one of the park's restaurants and savor delicious authentic cuisine. As evening approaches, you can unwind in the nearby hot spring area, immersing yourself in the unique charm of the volcanic region. | On the third day, you will visit Pearl Harbor, exploring this place steeped in historical memory. You can visit the USS Arizona Memorial, gaining an understanding of the historical significance of the Pearl Harbor attack. Afterwards, you can explore other museums and historical sites, delving deeper into the history of World War II. Come lunchtime, savor the local cuisine near Pearl Harbor, immersing yourself in the local culture. In the afternoon, take a leisurely stroll on the docks of Pearl Harbor, feeling the tranquil and solemn sea breeze. |

Table 2: Multi-day Trip Planning: Compared with ChatGPT, OKR-Agent presents more thorough consideration of elements.

Table 3 depicts the visual rendition of a fantastical adventure narrative, conceived by the OKR-Agent. Our method elaborated on the prompt: "one-paragraph background story of an NPC for the next Legend of Zelda game. The background story should mention ... by Jay Chou," resulting in the creation of four enriched scenes.

In this experiment, OKR-Agent demonstrated its prowess by producing narratives with richer and more layered content in comparison to SPP. The following is a narrative developed by OKR-Agent. Words highlighted in red represent the accurate responses to the input queries. Owing to space limitations, the comprehensive story has been relegated to the supplemental material:

| | |
|---|---|
| OKR-Agent | The camera reveals the mysterious and vast continent of Hyrule in the game world, with magnificent mountains, lush trees, and flowing streams in the distance. The protagonist confidently steps into Hyrule, while Ed Stark mysteriously emerges on the grass, his gaze unwaveringly fixed on the protagonist. A gentle piano melody resonates, gradually infused with Jay Chou's song "Slience", creating a tranquil and enigmatic atmosphere. Ed leads the protagonist through the dense forest, gradually guiding them towards a dark crevice. Ed swiftly navigates through the trees, displaying his combat skills and determination with the agility of a leopard. The incantation "Expecto Patronum" softly escapes from Ed's lips...The protagonist successfully defeats the forces of darkness, causing the obscurity to gradually dissipate, revealing the beautiful scenery and peaceful Hyrule. Ed and the protagonist stand side by side on the mountain peak, victorious expressions gracing their faces. The music reaches its climax, "Slience" resounds with a majestic movement that echoes through the sky, exemplifying the triumph of righteousness over evil. |
| SPP | In the mystical land of Hyrule, there lived a wise and powerful sorcerer named Eddard. Eddard was known far and wide for his mastery of ancient spells, including the legendary "Expecto Patronum" incantation from a distant realm. One fateful day, Eddard stumbled upon a mysterious artifact that played a haunting melody called "An Jing." The melody resonated with his soul, unlocking a hidden power within him. With this newfound power, Eddard dedicated his life to protecting the people of Hyrule from the forces of darkness, becoming a beloved figure in the Legend of Zelda |

Table 3: Trivia Creative Writing: OKR-Agent can produce richer and more layered content in comparison to SPP.

## 5 CONCLUSION

In this study, we demonstrate the effectiveness of OKR-Agent in optimizing task-solving pathways within intricate domains like storyboard generation and creative writing. By leveraging hierarchical structures and LLMs, it has enabled precise, coherent, and adaptive task planning and execution. The findings from the comparative experiments underscore OKR-Agent's superior global task planning and detail generation capabilities, offering substantial contributions to artificial intelligence research. It will be worth to investigate incorporating human-in-the-loop approaches to fine-tune and augment the capabilities of OKR-Agent, enabling real-time human interaction to enhance creative content generation and problem-solving processes.

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
