# OKR-Agent: An Object and Key Results Driven Agent System with Hierarchical Self-Collaboration and Self-Evaluation

# Appendix

## Short Video Storyboard Generation

### User Study:

**Dramatron**

| Plausibility of the Story | Text/Image Consistency | Image Continuity | Average |
|---|---|---|---|
| 4 | 3 | 2 | 3.5 |
| 4 | 3 | 4 | 3 |
| 3 | 2 | 2 | 3 |
| 3.5 | 2 | 2.5 | 2.5 |
| 4 | 3 | 2 | 3 |
| 3 | 3 | 2 | 3 |
| 3 | 2 | 3 | 3 |
| 4 | 4 | 3 | 3.5 |
| 4 | 3 | 3 | 3 |
| 4 | 3 | 2 | 3 |
| 1 | 2 | 1 | 1.5 |
| 2 | 3 | 2.5 | 2.5 |
| 3 | 3 | 2.5 | 2.5 |
| 4 | 2.5 | 2.5 | 3 |
| 2.5 | 3 | 2.5 | 2.5 |
| 4 | 2.5 | 2.5 | 3 |
| 3.5 | 3 | 3 | 3 |
| 3.5 | 3 | 2.5 | 3 |
| 3 | 2.5 | 2.5 | 3 |
| 3 | 2.5 | 2.5 | 3 |
| 4 | 3 | 3 | 3.5 |
| 3.5 | 3 | 3 | 3 |
| 4 | 3.5 | 3 | 3.5 |
| 3.5 | 3.5 | 3.5 | 3 |
| 3 | 3 | 2.5 | 3 |
| 3.5 | 4 | 3 | 3.5 |
| 4 | 3.5 | 3 | 3.5 |
| 3.5 | 3 | 2.5 | 3 |
| 4 | 3.5 | 3.5 | 3.5 |
| 4 | 3 | 2.5 | 3 |
| **Profession** 3.3 | 2.75 | 2.475 | 2.875 |
| **Amateur** 3.7 | 3.3 | 2.95 | 3.25 |
| **Weighted average** 3.4299 | 2.9304 | 2.6307 | 2.997 |

Dramatron Advantages & Weaknesses:
1. The story is rich, more like a script, lacking visual imagery.
2. The story is rich. There is a strong element of randomness in the visual content.
3. The plot twists in the story are too drastic.
4. The story is rich, but the plot twists are too drastic. There is a significant disparity between images and text.
5. The story is rich, but the plot twists are too drastic. The story development feels quite abrupt.
6. The story structure is rich, with significant changes in visuals.
7. The colors in the visuals are discontinuous.
8. The sudden involvement of parents in the story feels too abrupt. The composition of the visuals doesn't change much, and each frame feels too evenly distributed.
9. Overall, it's decent. The consistency in visuals is relatively weak. The plot twist feels too sudden.
10. The story is rich. There's a lack of consistency between text and visuals.
11. The story is good, but the ending feels abrupt.
12. Overall, it's okay. The plot is fine, but the consistency is weak.
13. The plot is intricate.
14. It's good. The plot is suitable for development.
15. The plot is intricate, and the visuals are rich.

**OKRAgent**

| Plausibility of the Story | Text/Image Consistency | Image Continuity | Average |
|---|---|---|---|
| 3 | 5 | 4 | 4 |
| 3 | 4 | 3 | 3 |
| 4 | 4 | 3 | 4 |
| 3 | 3 | 4 | 3.5 |
| 4.5 | 5 | 4 | 3 |
| 4 | 5 | 4 | 4 |
| 4 | 4 | 4 | 4 |
| 3 | 3 | 4 | 3.5 |
| 3 | 4 | 5 | 4 |
| 5 | 5 | 4 | 5 |
| 2 | 3 | 1 | 2 |
| 3 | 3 | 3 | 3 |
| 2.5 | 3.5 | 3 | 3 |
| 3 | 3 | 3 | 3 |
| 3 | 3.5 | 3 | 3 |
| 3 | 2.5 | 3 | 3 |
| 3 | 3.5 | 3.5 | 3.5 |
| 2.5 | 3 | 3 | 3 |
| 3 | 3 | 3 | 3 |
| 3 | 3.5 | 3 | 3 |
| 4 | 3 | 3 | 3.5 |
| 3 | 4 | 3.5 | 3.5 |
| 3.5 | 3.5 | 4 | 3.5 |
| 3 | 4 | 3.5 | 3.5 |
| 2.5 | 3.5 | 3.5 | 3 |
| 3 | 4 | 3 | 3.5 |
| 3.5 | 4 | 4 | 3.5 |
| 3.5 | 4 | 3.5 | 3.5 |
| 3.5 | 4 | 3.5 | 3.5 |
| 3 | 3.5 | 3.5 | 3.5 |
| **Profession** 3.236842105 | 3.605263158 | 3.342105263 | 3.342105263 |
| **Amateur** 3.25 | 3.75 | 3.5 | 3.45 |
| **Weighted average** 3.237986842 | 3.649855263 | 3.391342105 | 3.374692105 |

OKRAgent Advantages & Weaknesses:
1. The imagery in the text is sufficient, but the story seems childish.
2. The matching of visuals and text is good, but the timeline in scenes 4 and 5 is discontinuous.
3. Overall, it feels better than sample 1.
4. The continuity of visuals is good, like a story unfolding in a space.
5. The match between text and visuals is accurate, but scenes 4 and 5 are too repetitive. Everything else is okay.
6. The details of the child repairing the robot are accurate.
7. The alignment between visuals and text is quite high. The tones in different scenes harmonize well and appear very integrated. The story may seem a bit plain. The timeline in scenes 4 and 5 is discontinuous. Some parts of the visuals align well with the text description.
8. Overall, it's satisfactory but lacks standout points.
9. The plot is relatively continuous.
10. The plot is average, but the consistency in visuals is very good.
11. The visuals overall are quite good.
12. The story is not as good as the first one, but the visual continuity is good.
13. The plot is not as good as the first one, but the visual consistency is noticeably better.
14. The plot is touching, and the visuals give an overall warm feeling.
15. The content of the story and the visuals are harmonious. Overall, it's good.

## Another Story:

**User Input:**

Mom is making breakfast and calls for the boy to wake up. The boy lingers in bed. The puppy rushes in and wakes him up. The boy grabs his backpack and heads to school.

**Story:**

A pleasant morning unfolds before us, with sunlight streaming through the kitchen window, warm and gentle. The mother is fully engrossed in preparing breakfast. The camera, by chance, peers through a crack in the door, revealing the scene of the boy fast asleep, his sheets in disarray, and his pillow bearing traces of his hair. This tableau sketches a warm and familial ambiance. The kitchen is bustling with activity, while in contrast, the boy's room exudes a sense of tranquility

and serenity. The mother's busyness stands in stark contrast to the boy's quiet immersion in a sweet dream. She toils away in the kitchen, while the boy remains immersed in a beautiful reverie, laying the groundwork for the ensuing storyline. In terms of dialogue, the current scene doesn't feature specific exchanges or plot advancements. The focus lies in portraying a home filled with warmth and serenity through vivid descriptions of the surroundings and environment.

The mother walks from the kitchen to the boy's room. Gently, she knocks on the door, and with a warm yet firm voice, she awakens the boy. This call brings a touch of warmth to the room. Inside the boy's room, the sheets are in disarray, and he continues to sleep soundly. The mother is gentle yet resolute, while the boy is deeply immersed in his dreams.

The mother's knock awakens the boy, but he shows no response, remaining quietly in repose. Her call foreshadows the unfolding of the next events.

In their exchange, the mother addresses the child with a gentle yet firm tone, saying, "Child, it's time to wake up." Her voice is filled with love and determination, laying the foundation for the upcoming developments.

In the boy's room, the sheets are in disarray, and he remains deeply immersed in his dreams, showing no intention of getting up. Even though the mother intentionally acts as if she hasn't heard, the boy still lingers in bed. This creates a somewhat awkward atmosphere in the room. The boy appears somewhat lazy and innocent, while the mother wears a look of helplessness, her brows furrowed.

The boy continues to sleep, showing no sign of waking up, while the mother displays a hint of distress. The boy's reluctance to get out of bed propels the development of the ensuing events.

The mother calls out to the child again, saying, "Darling, aren't you going to get up?" There's a touch of resignation in her voice, once again foreshadowing the unfolding of the next events.

The mother's gaze falls on the nearby little dog, who lies obediently on the floor, wagging its tail and fixating on the food. Their interaction brings a touch of ease to the room. The little dog remains well-behaved, wagging its tail and focusing intently on the food. The mother sizes up the little dog, her eyes carrying a hint of cunning.

With a gentle wave of her hand, the little dog quickly springs to its feet, eagerly heading towards the boy's room. This action showcases the dog's vitality and loyalty. The dog's movement foreshadows the unfolding of the next events.

The mother says to the little dog, "Go on, wake up your master." Her voice carries a touch of cunning, while also revealing her anticipation for the dog's performance. This call from the mother propels the development of the ensuing events.

The little dog swiftly darted into the boy's room, leaping onto the bed without hesitation. It playfully frolicked on top of the boy, infusing the room with a lively atmosphere. The little dog continued to romp on the bed, ultimately rousing the boy. He appeared somewhat puzzled, unsure of what had just happened.

The boy slung his backpack over his shoulder, preparing for school. Inside his room, he swiftly organized his belongings and hoisted the backpack onto his back. The atmosphere in the room

grew tense, seemingly filled with both anticipation and apprehension. The boy seemed a bit hurried, while his mother watched from behind, her heart filled with contentment.

With quick, energetic movements, the boy tidied up and shouldered his backpack. He then waved goodbye to his mother. His actions brimmed with vitality and determination, signaling his imminent departure for school.

"Mom, I'm off to school," the boy declared. His voice resonated with energy and confidence, expressing his eagerness and assurance. This moment of parting also marked the conclusion of this scene.

**Story Visualization:**

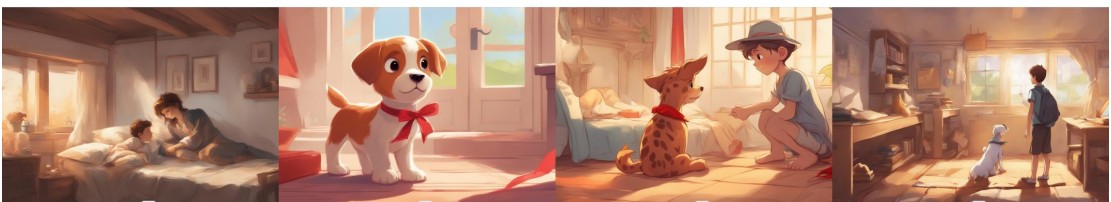

# Trivia Creative Writing

## Story Generation:

In this scene, the camera looks down from a distant height, revealing the vast continent of Hyrule. Distant mountains stand tall, lush trees dot the land, and a clear stream winds its way through. It is now the evening, with the gentle glow of the setting sun casting a serene and mysterious atmosphere over the continent.

The protagonist, Ed Stark, exudes an air of mystery and confidence as he strides into Hyrule from the edge of the grassy area. His gaze is resolute and sharp, conveying excitement and curiosity for the unknown world. While there is no overt interaction between the characters, the audience can feel the exchange and collision between the protagonist and the environment.

At this moment, Ed Stark doesn't speak, but his eyes reveal determination and passion. The background music seamlessly blends a soft piano melody with Jay Chou's song "Silence," creating a tranquil yet mysterious ambiance. The chirping of birds and the sound of flowing water artfully merge into the music, adding vividness and a touch of nature to the scene.

The scene begins from a high vantage point, showcasing a dense forest. The camera weaves through the trees, revealing the dimly lit surroundings, with thick foliage obscuring the sky. It is

now evening, and the lingering glow of the setting sun filters through the leaves, creating dappled patterns of light and shadow on the ground.

Ed swiftly wields his wand, moving nimbly between the trees. Each of his actions demonstrates exquisite combat skills and unwavering determination. His face carries a resolute expression, with furrowed brows and sharp eyes revealing his determination and intense focus.

The protagonist follows closely, attempting to keep pace with Ed's strides, but appears somewhat awkward in the dense forest.

Ed softly chants the incantation "Expecto Patronum," his voice gentle yet firm. A mysterious and exhilarating music gradually begins, emphasizing the tense yet hopeful atmosphere.

The rustling of leaves and distant insect chirping intertwine, creating an air of mystery in the forest.

The camera transitions from a long shot to a close-up, highlighting the twisted energy of the Dark Zone. This area is shrouded in heavy shadows, resembling a suspended dark cloud, pitch black and filled with an aura of terror and mystery. It is nighttime, and the bright moon hangs high above the Dark Zone, adding a touch of chilling elegance and mystique.

The protagonist and Ed grip their weapons tightly, maintaining a vigilant stance, preparing for an intense battle against the forces of darkness. The protagonist wields a radiant magical sword, with each strike releasing dazzling light. Ed's magic wand emits sparks of electricity, shattering the encroaching shadows. Their gaze is resolute, revealing a determination to combat evil. They remain focused and alert, driven by a pursuit of justice.

Amidst the fierce battle, they exchange dialogue:

Protagonist: "Ed, together we will conquer the darkness!"

Ed responds: "No problem! Our power will annihilate the malevolence!"

The music swells with passionate melodies, reaching crescendos alongside the protagonist and Ed's confrontation, adding depth to the battle.

The camera overlooks from a high vantage point, revealing the magnificent spectacle of the entire Hyrule. Beautiful and grand, Hyrule is characterized by undulating mountains, azure seas, and clear blue skies. It is noon, with the sun high in the sky, casting bright and warm rays upon everything, illuminating this beautiful land.

The protagonist stands tall, wielding their weapon, exuding confidence and determination in the wake of victory. They wear a smile, their eyes gleaming with joy and contentment.

Ed stands beside the protagonist, likewise displaying a joyful expression of triumph. The two stand shoulder to shoulder, radiating trust and friendship.

In this victorious land, the protagonist declares, "We did it! The forces of darkness have finally been defeated!" Ed responds, "Yes, we have triumphed over evil! This land is once again filled with peace and hope!"

The music is grand and solemn, embodying an atmosphere of victory and freedom. The sound effects are mighty, with instruments and voices gradually rising, symbolizing the immense power of justice. Natural sounds also intermingle, as bird songs and the sound of the waves intertwine, adding vibrancy to this moment of triumph.

## Story Visualization:

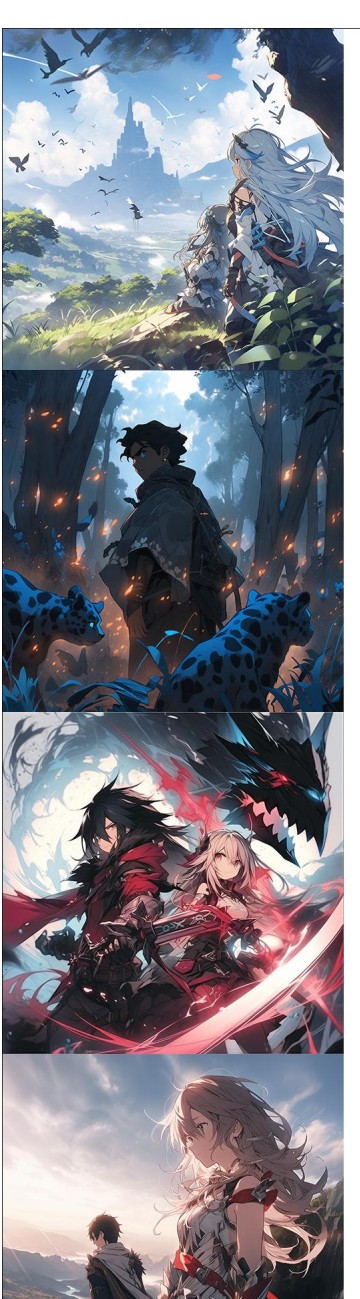

The camera reveals the mysterious and vast continent of Hyrule in the game world, with magnificent mountains, lush trees, and flowing streams in the distance. The protagonist confidently steps into Hyrule, while Ed Stark mysteriously emerges on the grass, his gaze unwaveringly fixed on the protagonist. A gentle piano melody resonates, gradually infused with Jay Chou's song "Slience", creating a tranquil and enigmatic atmosphere.

Ed leads the protagonist through the dense forest, gradually guiding them towards a dark crevice. Ed swiftly navigates through the trees, displaying his combat skills and determination with the agility of a leopard. The incantation "Expecto Patronum" softly escapes from Ed's lips.

The protagonist and Ed goto the dark realm, filled with twisted forms and unstable energy. Side by side, they fiercely combat the forces of darkness, closely cooperating and employing magic and weapons. The melody of "Slience" resurfaces, this time with the resonance of strings, expressing the heroes' determination and conviction against evil.

The protagonist successfully defeats the forces of darkness, causing the obscurity to gradually dissipate, revealing the beautiful scenery and peaceful Hyrule. Ed and the protagonist stand side by side on the mountain peak, victorious expressions gracing their faces. The music reaches its climax, "Slience" resounds with a majestic movement that echoes through the sky, exemplifying the triumph of righteousness over evil.