# OpenReview forum: "OKR-Agent: An Object and Key Results Driven Agent System with Hierarchical Self-Collaboration and Self-Evaluation"
_ICLR.cc/2024/Conference — ICLR 2024 Conference Withdrawn Submission_

### Official Review · Reviewer_YAvt · 2023-10-23

**Soundness:** 2 fair
**Presentation:** 3 good
**Contribution:** 1 poor
**Rating:** 3
**Confidence:** 4

**Summary:**

This paper proposes a hierarchical agent-based system called OKR-Agent for complex task solving. It utilizes a multi-level decomposition of objectives and key results along with role-specific agents and self-evaluations. Experiments on storyboard, trip planning, and creative writing tasks demonstrate enhanced performance over simple prompting.

**Strengths:**

The idea of hierarchical task decomposition and role assignment is logical and aligns well with real-world collaborative workflows.
Multi-level self-evaluations from different agent perspectives are interesting and help refine the solution.

**Weaknesses:**

- One of my major concerns is the limited technical novelty. Apart from the cute prompting, I did not see many method improvements from both technical and theoretical perspectives.

- Limited evaluation in a narrow domain with limited baselines. I would expect more complex open-world LLM-agent benchmarking results, and more language models to be tested. Currently, only simple comparisons with ChatGPT (not sure which model version as well) can not justify why the proposed prompting framework can be generalized to different models.

- More analysis is needed on the agent objectives, key results, and evaluation criteria to provide insights into what is being learned.

- Agent coordination details are lacking - how are dependencies managed?

- Safety mechanisms to prevent unbounded generation of objectives/agents need to be addressed.

- No comparison to hierarchical planning methods from classical AI is presented.

**Questions:**

- How does the performance scale with increasing levels of hierarchy? Is there a sweet spot?
- Can the approach work for very open-ended creative tasks without clear sub-objectives?
- Has the approach been tested on goal-driven tasks like process planning vs just content generation?
- What techniques are used to elicit useful evaluation criteria from the agents?
- How sensitive is performance to the quality of the initial user prompt?
- How difficult is it to construct such initial prompts from ambiguous and general tasks?

---

> ### Author Response · Authors · 2023-11-22
>
> Thanks for your comments, below are the feedback of your concerns:
>
> **Q1** One of my major concerns is the limited technical novelty. Apart from the cute prompting, I did not see many method improvements from both technical and theoretical perspectives.
>
> **A1** We want to emphasize that the proposed method is not a simple prompt engineering. In fact,  the complexity of the tasks we undertake exceeds the inference capabilities of existing Large Language Models (LLM). To avoid bottlenecks in LLM's performance, we have design the OKR based method, which allows the LLM to focus both on the global objectives and the details of the task. This approach enables the LLM to autonomously decompose tasks and evaluate the results of the decomposition, ensuring the accurate execution of target outcomes within the LLM's performance range.
>
>
> **Q2** Limited evaluation in a narrow domain with limited baselines. I would expect more complex open-world LLM-agent benchmarking results, and more language models to be tested. Currently, only simple comparisons with ChatGPT (not sure which model version as well) can not justify why the proposed prompting framework can be generalized to different models.
>
> **A2** Thanks for the suggestion and we will open-source our code and all experiments including intermediate results.  Our framework can work with other LLMs, the main difference is the stabilities of the generation caused by capacities of different LLMs.
>
>
> **Q3** More analysis is needed on the agent objectives, key results, and evaluation criteria to provide insights into what is being learned.
>
> **A3**  We offer additional ablation studies.
>
>
> **Q4** Agent coordination details are lacking - how are dependencies managed?
>
> **A4** The tasks for agents are automatically assigned by the Large Language Model (LLM) without any manual explicit assignment. The LLM autonomously decomposes the tasks based on their type and content, and allocates them to the appropriate agents.
>
> **Q5**  Safety mechanisms to prevent unbounded generation of objectives/agents need to be addressed.
>
> **A5** The generation of objectives and agents is bounded by the evaluation, where large number of objectives could lead to a low score in the evaluation. We also set the overall hierarchical levels to further prevent the possible unbounded generation.
>
> **Q6** How does the performance scale with increasing levels of hierarchy? Is there a sweet spot?
>
> **A6** Yes. Through ablation experiments, we compared the generation effects of 1-3 levels of hierarchy. Subjectively, three layers are currently the most suitable, with more than three levels showing no significant improvement in effectiveness.
>
> **Q7** Can the approach work for very open-ended creative tasks without clear sub-objectives?
>
> **A7** There are two methods that can have an impact: 1) By using the Few-Shot approach to inform the LLM of the required use cases. 2) By fine-tuning the LLM.
>
>
> **Q8** Has the approach been tested on goal-driven tasks like process planning vs just content generation?
>
> **A8** In our paper, "Multi-day Travel Planning" is an example of goal-driven tasks. Due to the length constraints of the paper, we did not include another task: "Wedding Planning." If permitted, we will supplement this in the accompanying materials.
>
> **Q9** What techniques are used to elicit useful evaluation criteria from the agents?
>
> **A9** We enable the LLM to customize assessment methods based on task objectives and assign them to different agents, thereby selecting the best inference results. The effectiveness would be further enhanced with the introduction of human interactive evaluation.
>
> **Q10** How sensitive is performance to the quality of the initial user prompt?  How difficult is it to construct such initial prompts from ambiguous and general tasks?
>
> **A10** We have adopted a method that allows the LLM to independently understand and decompose user input tasks (for example, providing the LLM with some cases to guide it in imitating methods that can successfully decompose objectives). Therefore, our algorithm is currently not sensitive to user input; the LLM will automatically analyze user objectives. Of course, more detailed descriptions lead to more effective results.

---

### Official Review · Reviewer_EcJ8 · 2023-11-01

**Soundness:** 3 good
**Presentation:** 2 fair
**Contribution:** 3 good
**Rating:** 6
**Confidence:** 4

**Summary:**

The paper presents the OKR-Agent, an advanced task-solving system that incorporates a hierarchical and self-collaborative approach to improve upon existing Large Language Model (LLM)-based methodologies. The system functions by breaking down tasks into Objectives and Key Results, allocating these to specific agents based on their roles within a workflow. This structure facilitates a more organized and coherent execution of tasks.

The contributions of the paper include:

1. Introducing a hierarchical and self-collaborative model for task decomposition and execution, which enables more structured and coherent task management.
2. Proposing a multi-level self-evaluation mechanism that allows agents to provide comprehensive evaluations from multiple perspectives, enhancing the accuracy and quality of outputs across strategic and executional levels.

The OKR-Agent was evaluated on tasks such as storyboard generation, creative writing, and trip planning, demonstrating superior performance in global task planning and detail generation. The results indicate that OKR-Agent is a significant step forward in the field of artificial intelligence, providing robust solutions for complex, multifaceted task-solving scenarios.

**Strengths:**

The paper tackles a compelling set of problems with its innovative integration of language agents and structured prompting, employing the Objectives and Key Results (OKR) framework to enhance task-solving capabilities. The concept is both novel and promising, as it fuses the flexibility of language-based AI agents with the goal-oriented precision of OKRs. This synergy enables the OKR-Agent to effectively decompose complex tasks into manageable sub-tasks, demonstrating a sophisticated approach to task execution. The idea of leveraging hierarchical structures to improve the clarity and efficiency of problem-solving is an excellent contribution to the field, showcasing a significant step forward in how AI systems can be structured for better performance and more coherent output.

**Weaknesses:**

Limited Comparative Benchmarking: The paper's comparison with existing methods seems constrained. A more extensive benchmark, including a variety of state-of-the-art systems, would better position the OKR-Agent's performance relative to the current landscape.

Heavy Reliance on Subjective Evaluation: The paper predominantly uses subjective evaluations for performance assessment. Objective metrics and statistical validation could provide a more balanced and reproducible evaluation framework.

Weakness in Evaluation Depth and Diversity: The paper lacks a diverse set of evaluation metrics. It primarily relies on subjective assessments, which, while valuable, may not capture the full performance spectrum of the OKR-Agent. Objective metrics like precision, recall, F1-score, or BLEU score for language tasks could provide a more comprehensive picture. The absence of these metrics is a significant weakness, as they are critical for substantiating the model's effectiveness across different scenarios and for different use cases.

Insufficient Ablation Studies: The paper does not present in-depth ablation studies. Ablation studies are critical for understanding which components of the proposed method contribute most significantly to its performance. By systematically removing or altering parts of the OKR-Agent, researchers can gain insights into the importance of each feature. This omission means that the reader has limited understanding of why the OKR-Agent works and which aspects are essential for its success.

Lack of Error Analysis: There is no detailed error analysis provided. An error analysis would help in understanding the limitations of the OKR-Agent and in which situations it might fail or underperform. This kind of analysis is crucial for iterative improvement and for setting realistic expectations for the model's deployment.

Weakness in Comparative Analysis Justification: The paper does not thoroughly justify the selection of systems used for comparative analysis. Understanding why certain systems were chosen as benchmarks and others were not is important for assessing the validity of the comparison. Without this justification, it's challenging to determine the relative standing of OKR-Agent in the broader field.

No Discussion on Model Robustness and Generalizability: The evaluation does not robustly test the generalizability of the OKR-Agent. It's unclear how well the agent would perform on tasks that were not part of the initial evaluation set, which is a significant weakness for claims of versatility and adaptability.

By addressing these weaknesses, the paper could strengthen its evaluation methodology and provide more insightful analysis into the of the proposed method.

**Questions:**

1. How does the OKR-Agent manage the computational complexity and resource allocation when scaling to more complex tasks and a larger number of agents?

2. In cases of ambiguous or poorly defined tasks, what mechanisms does the OKR-Agent use to ensure effective decomposition into Objectives and Key Results?

3. Could you provide insights into how the OKR-Agent might be adapted for use in highly specialized or technical domains that differ from those tested?

4. Are there any plans to conduct ablation studies to pinpoint the most critical components of the OKR-Agent's architecture for its task-solving performance?

5. How does the OKR-Agent interface with external data sources, and what strategies are implemented to ensure the integrity and applicability of this external data?

---

> ### Author Response · Authors · 2023-11-22
>
> Thanks for your comments, below are the feedback of your concerns:
>
> **Q1** Limited Comparative Benchmarking / Heavy Reliance on Subjective Evaluation / Lack of  Evaluation Depth and Diversity
>
> **A1** While we acknowledge the limited objective evaluations in our current work focusing on creative content generation, the assessment primarily relies on the subjective judgment of industry experts, which is a valid approach for this context. To align more closely with expert expectations, we have employed various engineering methods to enhance inference outcomes. These include using Few-Shot learning to improve the initial user input to OKR information conversion success rate, employing multi-round generation and assessment voting for optimal results, and integrating global and local information to enhance current inference outcomes. Additionally, our ablation studies on OKR generation outcomes further validate the effectiveness of our methods.
>
> **Q2** Insufficient Ablation Studies and Lack of Error Analysis
>
> **A2** We have conducted several ablation studies,  we will include them in main text or  the supplementary materials based on the page limits.
>
> **Q3** No Discussion on Model Robustness and Generalizability
>
> **A3** Due to the length constraints, we experimented two different types of experiments, including creative writing and goal planning in the submission. We will add more varied types of experiments if space allows.
>
> **Q4** How does the OKR-Agent manage the computational complexity and resource allocation when scaling to more complex tasks and a larger number of agents?
>
> **A4** We decompose OKRs into 3 hierarchical levels, and in hierarchical level, the objectives include 3-5 keys, which can address most tasks we tested.
>
> **Q5**  In cases of ambiguous or poorly defined tasks, what mechanisms does the OKR-Agent use to ensure effective decomposition into Objectives and Key Results?  Could you provide insights into how the OKR-Agent might be adapted for use in highly specialized or technical domains that differ from those tested?
>
> **A5** We have adopted the method of allowing the LLM to independently understand and decompose user input tasks (for example, providing the LLM with some cases to guide it to imitate methods that can successfully decompose objectives). For the key results generated by the Agent, we implemented "self-assessment" and "global assessment," hoping to enhance the effectiveness of the agent's execution.
>
> **Q6**  Are there any plans to conduct ablation studies to pinpoint the most critical components of the OKR-Agent's architecture for its task-solving performance?
>
> **A6**  We have added ablation experiments for the OKR generation stage. In these experiments, the results demonstrate that the customization of OKR objectives and the number of layers in OKR execution have a significant impact on the outcomes generated by the algorithm.
>
> **Q7**  How does the OKR-Agent interface with external data sources, and what strategies are implemented to ensure the integrity and applicability of this external data?
>
> **A7** There are two methods that can have an impact: 1) By using the Few-Shot approach to inform the LLM of the required use cases. 2) By fine-tuning the LLM.

---

> > ### Comment · Reviewer_EcJ8 · 2023-12-04
> >
> > Thank you for your feedback. After reviewing the other comments and the authors' rebuttal, I have taken into account the overall paper quality, and I have decided to maintain my original score.

---

### Official Review · Reviewer_ndXE · 2023-11-03

**Soundness:** 3 good
**Presentation:** 2 fair
**Contribution:** 2 fair
**Rating:** 5
**Confidence:** 3

**Summary:**

The paper proposes a way of using self-collaboration and self-correction mechanisms to enhance LLMs.  The idea is that in-depth knowledge is
required for complex tasks, so using specialized agents will increase performance.  Solving the task requires high-level strategic planning and low level execution.  The proposed framework uses a hierarchical generation of objects which are assigned to different role-specific agents and produces a multi-level collaborative evaluation. Experimental results on three different tasks show that the proposed approach outperforms other methods.

**Strengths:**

The main idea proposed seems novel and makes sense. It can provide a valuable paradigm for evaluation of complex tasks that are to be solved by LLMs.

**Weaknesses:**

The paper does not include enough details to be able to reproduce the work. For instance, the examples of the sentences used for generation prompt in Section 3.1 might not work with other LLMs or even with the same one after some time since LLMs get updated.

The examples shown do not use quantitative assessments. There are human subjects evaluations  but no indication of which differences are statistically significant, so saying the method proposed outperforms other methods does not seem to be strongly supported by the results presented.

**Questions:**

It is not clear to me that the method you proposed is easy to replicate.  Can you explain what is needed to replicate your results?

---

> ### Author Response · Authors · 2023-11-22
>
> Thanks for your comments, below are the feedback of your concerns:
>
> **Q1.** The paper does not include enough details to be able to reproduce the work ...
>
> **A1.**  Thanks for the suggestion and we will open-source our code and all experiments including intermediate results.  Our framework can work with other LLMs, the main difference is the stabilities of the generation caused by capacities of different LLMs.
>
> **Q2.** The examples shown do not use quantitative assessments ...%. There are human subjects evaluations but no indication of which differences are statistically significant, so saying the method proposed outperforms other methods does not seem to be strongly supported by the results presented.
>
> **A2.** Yes, our current primary focus is creative content generation. The evaluation of such tasks primarily relies on the subjective assessment of industry experts.  We will provide more ablation experiments on the OKR generation outcomes, which demonstrate their impact on the effectiveness of generation.

---

> > ### Comment · Reviewer_ndXE · 2023-11-22
> >
> > Thank you for answering my questions/comments.  I am glad to hear you will open-source your code.  At this point my evaluation of the paper is unchanged.

---

### Official Review · Reviewer_gKgL · 2023-11-05

**Soundness:** 2 fair
**Presentation:** 1 poor
**Contribution:** 3 good
**Rating:** 5
**Confidence:** 3

**Summary:**

In this paper, the authors propose a novel method to enhance the performance of LLMs in solving general, complex tasks. First, the main task is decomposed to subgoals (i.e. objectives); Then for each objective, a special agent (persona) and evaluation metrics are generated. In the solving stage, each objective specific agent will propose solutions and got reviewed by associated agents before an answer is accepted. The authors demonstrate that their proposed system can outperform DramaTron in storyboard generation tasks, and also can provide more detailed trip plans that ChatGPT for vacation plans.

**Strengths:**

1) The paper is novel in that it hierarchically decomposes the tasks into smaller objects and tackle these objectives with specialized agents.
2) The results seem impressive, i.e. that it can generate good stories that beat specialized models.

**Weaknesses:**

1) The biggest draw back of this paper is that, it is just not clearly written. I will be great to include code examples and intermediate solving results in the experiment session to help understanding.
2) If I understand correctly, for each key result the LLM has to be invoked at least multiple times during solution proposal and review. So overall this system seems costly and slow from an inference perspective. It will be great if the authors can compare with other methods from this perspective.

**Questions:**

Since the paper is not well written, I highly recommend the authors to consult with a professional proofreading service. Also, if will be great if the authors can help clarify the following questions in the next version:

1) The proposed system will decompose an task into many hierarchies. So who is deciding the number of hierarchies needed? From algorithm 1 it looks like an hyper parameter.
2) Since the task will be broken into many objectives, are there causal or temporal links between different objectives? For example, how objective 2 can be solved should depend on objective 1's solution.
3) If so, how can the different agents coordinate to solve the objectives in the correct order? From algorithm 2 it seems that each agent is just independently updating a portion of the final answer.
4) What if one agent cannot give a satisfying answer? What will happen to the whole task?
5) Are there any "replannings" where certain objectives should be removed/reconsidered?

---

> ### Author Response · Authors · 2023-11-22
>
> Thanks for your comments, below are the feedback of your concerns:
>
> **Q1:**  The biggest draw back of this paper is that, it is just not clearly written. I will be great to include code examples and intermediate solving results in the experiment session to help understanding.
>
> **A1:**  Thanks for the suggestion and we will open-source our code and all experiments including intermediate results.
>
>
> **Q2:**  If I understand correctly, for each key result the LLM has to be invoked at least multiple times during solution proposal and review. So overall this system seems costly and slow from an inference perspective. It will be great if the authors can compare with other methods from this perspective.
>
> **A2:**   Yes, there is always a trade-off between the cost of inference and the overall quality of solutions. Our inference process is divided into two stages: OKR generation and Agent working stage. In the OKR generation phase, well-designed prompts enable the LLM to effectively improve the OKR inference results, generating reasonable keywords. This process relies on the model's inference capability rather than content generation capacity, so it does not require excessive Token consumption. In the Agent working stage, the overall cost is directly proportional to the number of tokens. Here, we validated several different LLM base models (including gpt-3.5-turbo, gpt-4, flan-alpaca-xl-3B, etc.) under the constraints of OKRs, all capable of producing satisfactory content generation results. This observation also helps in reducing generation costs. Based on GPT-API's pricing, the cost for a single creative writing generation is about \$0.5 to \$0.6, and even lower, less than \$0.1, if using local models.
>
> **Q3:**  The proposed system will decompose an task into many hierarchies. So who is deciding the number of hierarchies needed? From algorithm 1 it looks like an hyper parameter.
>
> **A3:**  We set the number of hierarchies empirically. Based on our experiments, 3 layers are sufficient for the tasks we tested and there is no significant improvement with more than 3 layers. We will provide this ablation in the revision.
>
>
> **Q4:**  Since the task will be broken into many objectives, are there causal or temporal links between different objectives? For example, how objective 2 can be solved should depend on objective 1's solution.    If so, how can the different agents coordinate to solve the objectives in the correct order? From algorithm 2 it seems that each agent is just independently updating a portion of the final answer.
>
> **A4:**  There is no explicit link between objects although there could be shared components of task for different objects. Both the tasks and agents are automatically assigned by the Large Language Model (LLM) without any manual assignment.
>
> **Q5:**  What if one agent cannot give a satisfying answer? What will happen to the whole task?  Are there any "replannings" where certain objectives should be removed/reconsidered?
>
> **A5:** Once OKRs are generated, they will still fixed. We use an assessment and adjustment process during the OKRs creation stage to ensure the OKRs are correct and necessary. After an agent completes the execution of OKRs, the outcomes produced will be "evaluated" and "revised" again according to the OKR objectives.

---

### Author Response · Authors · 2023-11-21
**feedback w.r.t details and evaluations**

We thank the reviewers for their valuable comments and recognition of the results and novelty of OKR-Agent. Below we address the major concerns and we will incorporate the remaining suggestions in the revision. Our code is ready to release.

## details for reproduce.
Thanks for the suggestion and we will open-source our code and all experiments including intermediate results.

## evaluations and ablations.
Thank you for your suggestion. Due to the length constraints of the submission, not all experiments were included in the paper. However, we have added the results of "tasks in more domains" and "ablation studies" in the supplementary material to fully address your concerns. In our experiments, we also included two "goal-driven tasks": one is "Travel Planning" and the other is "Wedding Planning." Both achieved satisfactory generative results. In the ablation studies, we conducted several sets of experiments, including "ablation experiments for the OKR generation method" and "ablation experiments on the number of keys and hierarchical levels in OKRs."

---

### Meta-Review · Area_Chair_eXpC · 2023-12-05

**Metareview:**

This paper presents a novel method for improving the performance of LLMs in complex tasks. The proposed approach decomposes the main task into subgoals assigned to objective-specific agents, with a multi-level self-evaluation mechanism, allowing each agent to offer evaluations from all related perspectives. The proposed approach was evaluated on tasks such as storyboard generation, creative writing, and trip planning. In general, most of the reviewers appreciate the novelty and strong empirical results on a compelling set of benchmarks. However, most of them also have concerns about the writing quality and clarity of this paper, as many details appear to be missing. After rebuttal and revision, the reviewers have still not been fully convinced of the publication readiness. Thus, I recommend rejecting this paper. The authors are encouraged to incorporate the feedback and resubmit in a future venue.

**Justification For Why Not Higher Score:**

Insufficient quality and clarity of writing.

**Justification For Why Not Lower Score:**

N/A

---

### Decision · Program_Chairs · 2024-01-16

Reject